# Community engagement in Indigenous food systems contamination studies: A systematic scoping review

Federico Andrade-Rivas[1,2,3], Hallah Kassem[4], Kira Mok[5], Chenoa Cassidy-Matthews[6], Matthew Little[1], Mélanie Lemire[7,8], Annalee Yassi[9], Jerry Spiegel[9]*

1 School of Public Health and Social Policy, University of Victoria, Victoria, British Columbia, Canada, 2 Instituto de Salud y Ambiente, Universidad El Bosque, Bogotá, Colombia, 3 Geography Department, University of Colorado Boulder, Boulder, Colorado, United States of America, 4 Environmental and Occupational Health Sciences, University of Washington, Seattle, Washington, United States of America, 5 Department of Sociology and Anthropology and Department of Health Sciences, Northeastern University, Boston, Massachusetts, United States of America, 6 Indigenous Health Research Unit, Vancouver Coastal Health, Vancouver, British Columbia, Canada, 7 Axe santé des populations et pratiques optimales en Santé, Centre de Recherche du CHU de Québec, Hôpital du Saint-Sacrement, Chemin Sainte-Foy, Québec, Canada, 8 Département de Médecine Sociale et Préventive, Université Laval, Pavillon Ferdinand-Vandry, Québec, Canada, 9 School of Population and Public Health, The University of British Columbia, Vancouver, British Columbia, Canada

* jerry.spiegel@ubc.ca

## Abstract

### Introduction

Indigenous food systems are vital for maintaining cultural practices, physical and mental well-being, and community health. However, these systems are increasingly threatened by environmental contamination, exacerbating health disparities. Despite growing recognition of the importance of Indigenous knowledge in environmental health research, there is limited systematic evidence on how well community engagement is incorporated into studies investigating contamination of Indigenous food systems. This scoping review aims to assess reported practices for engaging Indigenous Peoples and the use of study results to support community-driven initiatives.

### Methods

A systematic scoping review was conducted on peer-reviewed articles published between January 2010 and July 2024 that assessed contamination in Indigenous food systems with a human health dimension. The search included three databases: Web of Science, Scopus, and CAB Direct, yielding 2,203 articles. After applying inclusion and exclusion criteria, 202 studies were retained for final analysis. Data were extracted on study characteristics, community engagement strategies, Indigenous knowledge integration, and reported use of study results by Indigenous Peoples. The analysis was conducted using the PRISMA framework.

**Data availability statement:** All relevant data are within the manuscript and its Supporting Information files.

**Funding:** The author(s) received no specific funding for this work.

**Competing interests:** The authors have declared that no competing interests exist.

## Results

Most studies (97%) employed quantitative methods, with fewer incorporating qualitative or mixed-method approaches. While community engagement was mentioned in about two-thirds of the studies, the depth of engagement varied significantly. A quarter of studies included Indigenous authors and only a small proportion reported meaningful collaboration with Indigenous Peoples throughout the research process. Studies with Indigenous authorship were more likely to report community engagement activities and utilization of results for broader community initiatives.

## Conclusion

The increasing recognition of Indigenous and traditional knowledge within academia must extend beyond intellectual discourse to address health disparities. Indigenous Peoples have long advocated for self-determination and engagement in research conducted in their communities. As part of broader reconciliation efforts with Indigenous Peoples the environmental health scientific community must reciprocate these efforts by integrating discussions into scientific literature about community participation and implementation of study results. This review highlights the need for robust and meaningful community engagement in environmental health research related to Indigenous food systems.

## Introduction

Pollution is the world's leading environmental risk factor, claiming nine million deaths per year, triple the number of deaths caused by HIV/AIDS, malaria, and tuberculosis combined [1–3]. However, there is considerable uncertainty regarding the multiple effects of chemical pollution exposure on human health, particularly among historically marginalized populations [2–5]. The growing complexity of environmental contamination, including emerging chemicals, mixtures, and multi-level drivers of pollution, requires solutions beyond conventional risk management approaches based on source-by-source and pollutant-by-pollutant assessments. In addition to more comprehensive assessments of health effects, the inclusion of affected populations as key actors in remedying the circumstances that render them victims of contamination is also emphatically necessary.

Researchers are increasingly including multi-level and system-oriented approaches to assess and identify structural solutions to the globalized pollution crisis [6]. This is the case for scholars involved in the emergent planetary health field, which has seen rapid growth in the last decade [7]. The need for diverse conceptual frameworks that expand our current approaches in environmental health and sustainability research has increased recognition of the importance of the traditional knowledge of Indigenous Peoples [8–10]. This stems from the fact that Indigenous traditional knowledge is, by definition, relational, holistic, and focused on the interconnection of the determinants of human and planetary health [9,11].

However, recognition of Indigenous Peoples knowledge to the global community is often merely symbolic, and the insights that traditional knowledge contribute to the pursuit of healthier and sustainable solutions have yet to be reflected in most planetary health research [9,12,13]. One ongoing challenge is the limited engagement of researchers with Indigenous Peoples, even when studies are conducted with Indigenous participants or in their territories [14]. Without actions that promote community engagement and acknowledge communities' goals for well-being improvement, researchers may struggle to create bridges between diverse ways of knowing and being. This is particularly evident in environmental health research in the Circumpolar North, where scholars may not fully recognize the changing political climate in the region where research is viewed as an act of self-determination to promote Indigenous Peoples well-being [15]. The limited inclusion of Indigenous voices in published research has led to some specific initiatives from the academic community. For example, recent updates to three rural health journals' editorial standards require that research about Indigenous Peoples include Indigenous Peoples as authors or provide evidence of a participatory process of Indigenous community engagement [16].

Although interest in fostering meaningful collaborations with Indigenous Peoples is increasing, Indigenous Peoples continue to experience greater disease burdens in comparison to the general population [17–19]. In a study conducted across 23 countries, Anderson and colleagues compared the health of Indigenous Peoples to that of non-Indigenous populations and documented poorer health outcomes, such as reduced life expectancy at birth, and higher maternal and infant mortality [20]. The multi-factorial social determinants of health that explain these differences are intertwined with the toxic exposures associated with the uneven distribution of pollution.

Most pollution-related deaths occur in low- and middle-income countries, where Indigenous Peoples face high risk of contamination exposure and disease development [1–4]. However, Indigenous Peoples in high-income settler colonial countries such as Canada and the USA are also disproportionally threatened by chemical exposures, where many communities have drinking water advisories, unsafe dwellings, live near polluted industrial areas, or are impacted by the long-range transport of persistent pollutants [21–26]. Moreover, despite the advances in environmental health research under the environmental justice and health equity frameworks, there is a persistent lack of chemical exposure research involving Indigenous Peoples [27].

Indigenous food systems and traditional food harvesting remain vital for maintaining cultural protocol, governance structures, intergenerational knowledge transmission, physical activity, collective mental health, and community well-being [28–32]. However, the effects of colonization, including unhealthy Western diets, exposure to toxicants, and systemic racism, have impacted Indigenous Peoples' access to healthy traditional foods [33–38].

Amidst efforts of various Indigenous Peoples to reconstitute traditional healthy practices, it is crucial to consider that traditional food sources may present a variety of toxic chemicals through local or foreign contamination [39–42]. For example, some local sources of exposure to toxic chemicals for Indigenous Peoples in the Arctic region come from ammunition (i.e., lead) and cigarette consumption (i.e., cadmium) [40]. However, the long-range transport of contaminants from urban, industrial and agricultural sources in southern latitudes are major sources of contamination in the Arctic, which can bioaccumulate and biomagnify in wildlife species consumed by Indigenous Peoples [24,43,44]. Thus, despite the absence of local sources of persistent organic pollutants and mercury, exposure continues to be higher for Inuit communities than the general population [40]. A large number of communities in the Circumpolar region have cultural practices deeply connected to the oceans and rivers, including the traditional consumption of marine mammals and other wild foods, which may elevate their risk to these toxic exposures [40,45]. Toxic exposures through wild foods consumption have also been studied among other Indigenous Peoples, such as populations in the Amazon River basin [46,47].

The complex challenge of chemical exposure through traditional food systems warrants adopting risk assessment approaches that meaningfully engage with Indigenous Peoples. Such engagement is necessary to understand and manage the unique risk context within their communities while considering the benefits of their preferred locally harvested foods [14]. Community engagement has the potential to strengthen the relevance and rigour of research on human

well-being [48–53]. Moreover, the collaboration between communities and researchers has the potential to forge alliances to advocate for regulating or banning toxic chemicals that protect the safety of traditional foods, while promoting broader positive impacts. This was the case of the role played by Inuit leadership from Nunavik in collaboration with researchers to advocate for banning polychlorinated biphenyls (PCBs) and organochlorines (Ocs) [51].

Consequently, emerging approaches in planetary health and the established environmental justice movement have called for environmental health and exposure science to promote community engagement in the research process [9,13,54]. Indigenous scholars have strongly advocated for environmental health research that is not solely centred on documenting damage but also uplifts communities' desires, aspirations, objectives, wisdom, and hope [55]. However, there is limited systematic evidence on how fields such as toxicology, exposure science, and environmental health research are responding to these calls. A 2018 systematic review of environmental health research in the Circumpolar North concluded that it is not a common practice among researchers to report on community engagement and their relationship with Indigenous partners [15]. Attention to how the global scientific community is reporting on its community engagement strategies and how Indigenous Peoples are making use of study results is needed. This is particularly warranted in contamination studies on traditional food systems where biomonitoring of human, environmental, and food samples may pose additional ethical challenges [56,57].

It is especially timely to explore the progress of the scientific community in upholding the self-determination of Indigenous Peoples in the context of ongoing development of integrative approaches to human health (e.g., planetary health) that emphasize in their discourse the importance of articulating Indigenous knowledge. We contribute to filling this gap by conducting a systematic scoping review of empirical research articles that study contamination of Indigenous traditional food systems. This review aims to identify to what extent researchers report both the engagement of communities in the research process and how communities make use of study results.

## Methods

The search was limited to research articles on the contamination of Indigenous food systems that included a human health dimension in the data collection or analyses. Our research strategy aimed to capture a broad group of Indigenous food systems, Indigenous Peoples, and methodologies used to assess the potential association between food contamination and human health (Table 1). Thus, we did not restrict the type of methods used, food sources analyzed, study location, or the language of the study. To be included in the review, research articles needed to empirically assess chemical contamination, directly or indirectly, and discuss potential or measured impacts on human health or well-being. Moreover, included studies had to assess contamination of Indigenous food systems, and be conducted in an Indigenous territory or among Indigenous Peoples. We searched for research articles published between January 2010 and July 29, 2024 (the day when the search concluded) in three databases, each containing several collections (Web of Science, Scopus, CAB direct and global). We did not examine cited or citing references, nor did we search grey literature or consult external sources.

The search process and results were documented following the Preferred Reporting Items for Systematic Reviews (PRISMA) approach [58–60]. To identify studies that met our inclusion criteria we screened the articles in two phases. First, we screened the title and abstract and retained articles that met the inclusion criteria and articles that presented insufficient information at this stage to determine eligibility. Second, we conducted a full-text screening of the articles retained in the previous phase. We then reviewed the full manuscripts of the retained articles and extracted the data using a structured data collection form developed by the authors. Each article was screened and reviewed independently by two co-authors, and disagreements in the data extracted were resolved by consensus reached through discussion. To promote consistency, the first author screened and reviewed all the articles, while HK screened and reviewed articles published between 2010 and 2022 and KM between 2023 and 2024. We used the Covidence web-based collaboration software platform to identify duplicates, screen articles, identify conflicts to be resolved, develop the data extraction form, and extract data [61].

**Table 1. Scoping review search strategy and exclusion criteria used to select publications.**

| Search Strategy Component | |
|---|---|
| Electronic databases | Web of Science<br>Scopus (Includes MEDLINE and EMBASE)<br>CAB direct and Global Health |
| Search terms | (food* OR fish* OR hunt* OR gather*)<br>AND<br>(indigen* OR aborig* OR native* OR amerind* OR "first nation*" OR "first people*")<br>AND<br>(contamin* OR pollut* OR "environmental health" OR toxic* OR "exposure assessment") AND<br>(health OR "human health" OR wellbeing OR well-being) |
| Fields searched | Topic, abstract, title, or keywords. |
| Type of documents | Research Articles |
| Time | Published between January 1, 2010 and July 29, 2024 (the day of the literature review) |
| Languages | No restriction |
| Exclusion criteria | • Duplicates<br>• Unrelated topic (e.g., no contamination of food systems, no Indigenous Peoples)<br>• Different document type (e.g., review, methodological article, commentary)<br>• Unavailable documents |

* The asterisk is used to include all the words that have the same root but multiple endings (e.g., hunt and hunting).

The data extracting form included general information about the article (e.g., year, title), the Indigenous Peoples name provided by the authors, the country of the study, and the methods used. In addition, we captured activities and strategies for community engagement in the research process and the community use of the study results, as reported in the publication, including the acknowledgement and contribution recognition sections. Finally, we reviewed author affiliations to identify links to Indigenous Peoples organizations or institutions. We verified whether the organizations were governed by Indigenous Peoples and classified those that explicitly mention their connection to an Indigenous Nation as "Indigenous affiliation".

Data cleaning, calculation of descriptive statistics and proportions, and graphs were conducted in the R statistical computing environment (version 4.1.1), and the code was executed in RStudio (version 1.4.1717) [62,63]. The review protocol was developed by the research team with the support of a librarian with public and environmental health expertise (protocol was not registered prior to publication).

## Results

Our search in the three databases yielded 2,203 items, among which 929 were duplicates (Fig 1). After conducting title and abstract screening, 978 items were removed for not meeting the inclusion criteria. We conducted a full-text screening with the remaining 296 studies and identified 94 additional studies that did not meet our criteria. Some of the main reasons for excluding research articles were: not being conducted on Indigenous food systems or populations; not assessing exposure to contamination; and not evaluating exposure related to food consumption. We retained 202 studies for the final review (complete reference list and data extracted for each article are provided in S1 and S2 Tables).

We initially identified 189 unique Indigenous Peoples in the participating studies. This count reflects the names used by authors in the original studies (S3 Table). However, some of the names used by authors may correspond to the same Indigenous Peoples or are broader terms that aggregate more than one population (e.g., First Nation, Inuit). We further merged Indigenous Peoples into 17 distinct groups (Table 2).

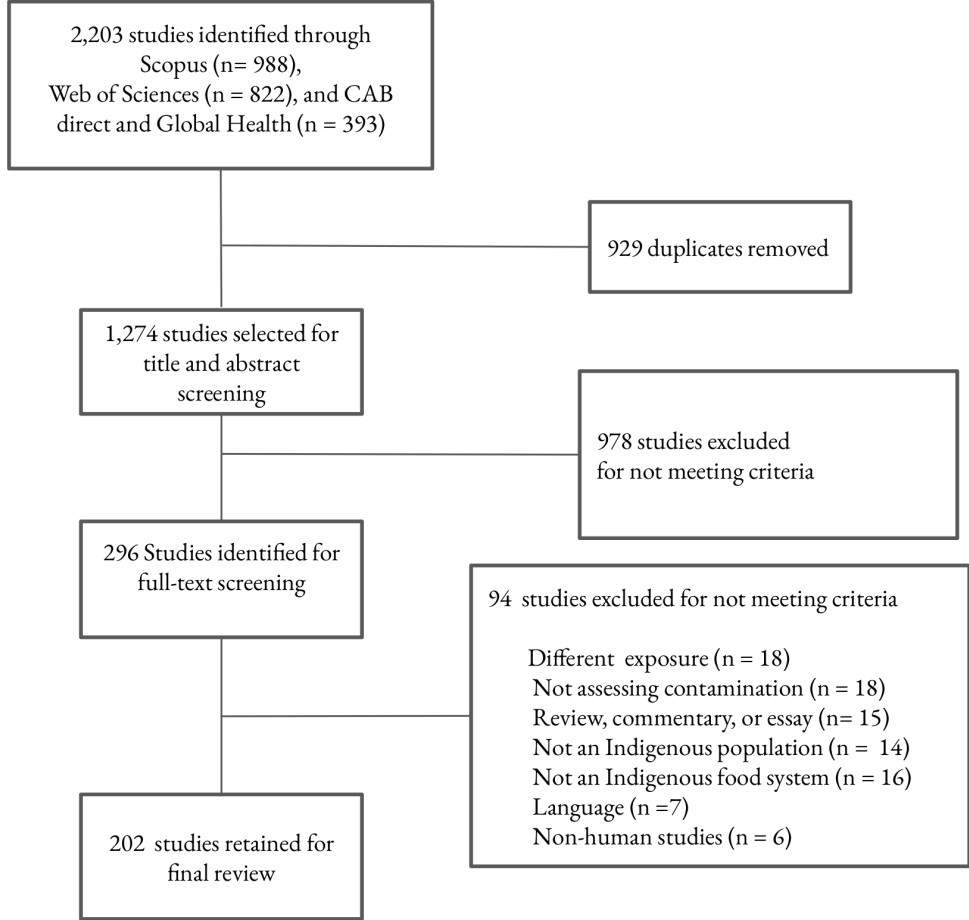

**Fig 1. Preferred Reporting Items for Systematic Reviews (PRISMA) flow diagram detailing the article selection process.**

Studies were conducted in 22 countries from North America, South America, Eurasia, Oceania, and Africa, with five conducted across two countries. However, the vast majority of studies were located in North America. About half of the studies were conducted in Canada, followed by the United States, Brazil, Peru, and Russia. Only four studies were conducted in Africa or Asia (Table 2). Countries in the Americas are disproportionately represented in our sample, with no apparent correlation between the number of studies per country and their total Indigenous population. For example, no studies were conducted in Mexico or Guatemala, which have the largest number and proportion of Indigenous Peoples population in the Americas [64], while Canada accounted for more than half of all the studies included in the review. This suggests that more studies need to be conducted and published from countries with large Indigenous populations that are no doubt facing considerable environmental concerns.

Studies included in our review used a broad range of methods to assess contamination and human health associated with Indigenous food systems (Table 3). Of the 202 studies, 143 used multiple methods, and the remaining 59 used a single method. About 97% of the studies included at least one quantitative method (e.g., survey analysis, biomonitoring), ~10% combined quantitative and qualitative methods, and ~3% were exclusively qualitative. Overall, 184 studies directly measured contamination in environmental, food, or human samples such as blood and urine. Sixteen studies that assessed contamination through biological samples also included a qualitative component. In addition, out of the 14

**Table 2. Countries and Indigenous Peoples groups where studies were conducted.**

| Country (Number of studies)* | Indigenous Peoples group | Number of studies* |
|---|---|---|
| Canada (n = 101) | Circumpolar | 31 |
| | First Nations | 75 |
| | Metis | 9 |
| United States (n = 36) | Alaska Native | 2 |
| | Circumpolar | 11 |
| | Hawaiian | 1 |
| | Native American | 23 |
| Brazil (n = 16) | Amazon Indigenous Peoples | 15 |
| | Not specified | 1 |
| Peru (n = 12) | Amazon Indigenous Peoples | 12 |
| Russia (n = 11) | Circumpolar | 11 |
| Bolivia (n = 4) | Amazon Indigenous Peoples | 3 |
| | Andean Indigenous Peoples | 1 |
| New Zealand (n = 4) | Maori | 1 |
| | Pacific Islander | 3 |
| Colombia (n = 3) | Amazon Indigenous Peoples | 3 |
| Greenland (n = 3) | Circumpolar | 3 |
| India (n = 3) | Tribal people of Koraput | 1 |
| | Tribes of Assam | 1 |
| | Not specified | 1 |
| Australia (n = 2) | Aboriginal and Torres Strait Islander | 2 |
| Chile (n = 1) | Mapuche | 1 |
| Democratic Republic of the Congo (n = 1) | Not specified | 1 |
| Ecuador (n = 1) | Amazon Indigenous Peoples | 1 |
| French Guiana (n = 1) | Amazon Indigenous Peoples | 1 |
| Ghana (n = 1) | Not specified | 1 |
| Guyana (n = 1) | Amazon Indigenous Peoples | 1 |
| South Africa (n = 1) | Bapedi | 1 |
| Suriname (n = 1) | Amazon Indigenous Peoples | 1 |
| Swaziland (n = 1) | Swazi | 1 |
| Taiwan (n = 1) | Amis | 1 |

*Studies conducted across two countries and more than one Indigenous Peoples are repeated. Thus, the total number of studies in this table is greater than the number of articles included in the review.

**Table 3. Methods, community engagement, and use of results categories description. Number of publications classified in each category.**

| Category | Description | N* |
|---|---|---|
| *Methods* | | |
| Quantitative Survey analysis | Analysis of data collected through surveys or questionnaires analyzed using quantitative research strategies. For example, exposure assessed using self-reported eating practices and frequencies | 105 |
| Biomonitoring human samples | Exposure assessed measuring contamination levels in human biological samples such as blood, hair, or urine | 94 |
| Biomonitoring food samples | Exposure assessed measuring contamination levels in animal or plant species that are part of Indigenous food systems | 89 |
| Environmental monitoring | Exposure assessed measuring contamination levels in the environment. For example, measuring contamination in water where fish are harvested, or plants that feed mammals that are hunted | 33 |
| Simulation or modelling | Exposure assessment based on secondary or historical data to model or simulate contamination levels. All studies categorized under this category where also classified as "Secondary data analysis" | 11 |
| Qualitative | Analysis of data collected through qualitative methods such as ethnography, participatory observation, interviews, and focus groups. For example studies to identify practices leading to exposure, or risk perception of traditional food consumption | 27 |
| Indigenous knowledge integrated to Western science methods | Indigenous concepts, categories, and knowledge integrated in the study designing. For example, identifying sampling areas, species, and seasons for data collection. | 15 |
| Indigenous Methods | Although they have been practised for many generations, these methods are recently emerging in scientific publications. Indigenous methods may be context-specific; however, they usually include strategies beyond data collection and are intended to build relationships and foster self-determination. For example, storytelling and talking circles. | 2 |
| Secondary data analysis | Analysis of secondary data. This category was paired with the specific type of method used in the studies. For example, a study using secondary data from contamination levels in blood, would be categorized both in this category and "Biomonitoring human samples" | 67 |
| *Community Engagement* | | |
| Study requested by community | Indigenous Peoples member or the community contacted the researchers to collaborate in the project | 35 |
| Results discussion in collaboration with community | Indigenous member were part of the discussion of the results and conclusions of the study | 31 |
| Research methods designed in collaboration with community | Indigenous members collaborated in the research design, for example sharing knowledge to design the sampling strategy | 48 |
| Community members participated conducting research activities | Indigenous members conducted or assisted data collection | 100 |
| Activity to receive feedback from community | Explicit activity to receive community feedback on the study results, methods, or conclusions. For example, a workshop to validate study results | 72 |
| Information session | Information about the study was provided to the community through group events or other communication strategies. This category includes activities or strategies intended to inform but not explicitly to receive feedback. | 49 |

*(Continued)*

**Table 3.** (Continued)

| Category | Description | N* |
|---|---|---|
| Not mentioned | – | 75 |
| *Community Use* | | |
| Study results are already part of a broader strategy | Authors provided information on how the study findings or research process are articulated with strategies that are currently taking place. The role of the study as part of a broad strategy is discussed | 27 |
| Study results are expected to inform an specific broader strategy | Authors provided information on how the study findings or research process will be articulated with strategies and other projects. The role of the study in this broader picture is discussed. We did not include studies that did not mention an explicit program, objective, or following steps that will be conducted. | 47 |
| Not mentioned | – | 128 |

*For the "methods" and "community engagement" categories more than one level can be assigned to a single study. Thus, the sum is larger than the number of articles included in the review.

studies that included Indigenous knowledge, six were exclusively quantitative, and eight included a quantitative and a qualitative component.

Only two studies reported using Indigenous research methods in addition to Western qualitative and quantitative methods. Briefly, Cott and colleagues collaborated with Indigenous experts to evaluate appearance-based traditional methods to determine food quality and contamination levels of fish consumed by local communities (Gwich'in Settlement Area, NWT, Canada) [65], while Lucier and collaborators applied storytelling methods to assess the multiple impacts of water quality degradation in an Anishinaabe First Nations community (Ontario, Canada) [66].

We also examined reported community engagement activities and information on community use of the study results. About a third of all articles did not mention any strategy for community engagement. The most common activity linked to community engagement was the participation of community members in conducting or assisting in data collection (Table 3). Another common strategy was to implement different activities to receive feedback from community members regarding the study results or research process. For example, researchers received feedback through community workshops or from a community committee designated to represent the Indigenous population. In addition, 23% of studies reported activities to include the Indigenous Peoples in designing the methodology, and 15% had strategies in place to include community members' perspectives on the study results and discussion. Moreover, 17% of the studies were conducted following an invitation from an Indigenous population.

All studies articulating Indigenous and Western knowledge reported at least one activity associated with community engagement (Fig 2). In addition, a relatively large proportion of qualitative studies (94%) mentioned at least one community engagement strategy. Studies incorporating environmental monitoring had the highest proportion of not reporting any community engagement strategy (25%), which was higher than for studies including the analysis of human or food samples (16% and 18%, respectively). The proportion of studies reporting Indigenous community members conducting research activities did not show large differences across research methods, with the largest for studies including food biomonitoring (26%) and the lowest for environmental monitoring (20%). Twenty-two percent of studies using survey methods reported that community members conducted research activities. However, out of the 11 studies that used surveys but did not conduct any biological sampling, only three included community members in research activities.

Approximately 60% of articles did not report on community use of study results for current or future projects, interventions, or programs (Table 3). When disaggregating by study methods, articles documenting the inclusion of a qualitative component or approaches including Indigenous knowledge reported a relatively higher proportion of current or expected

A.

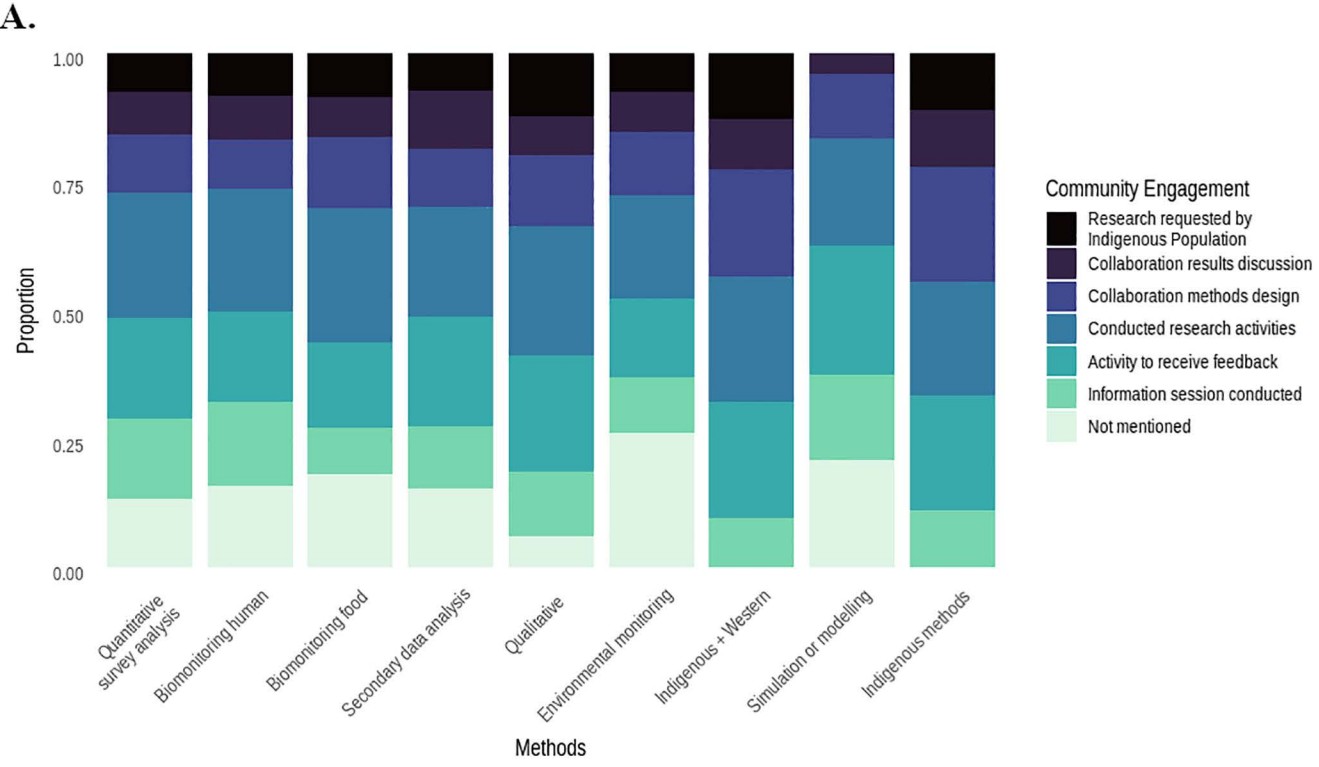

B.

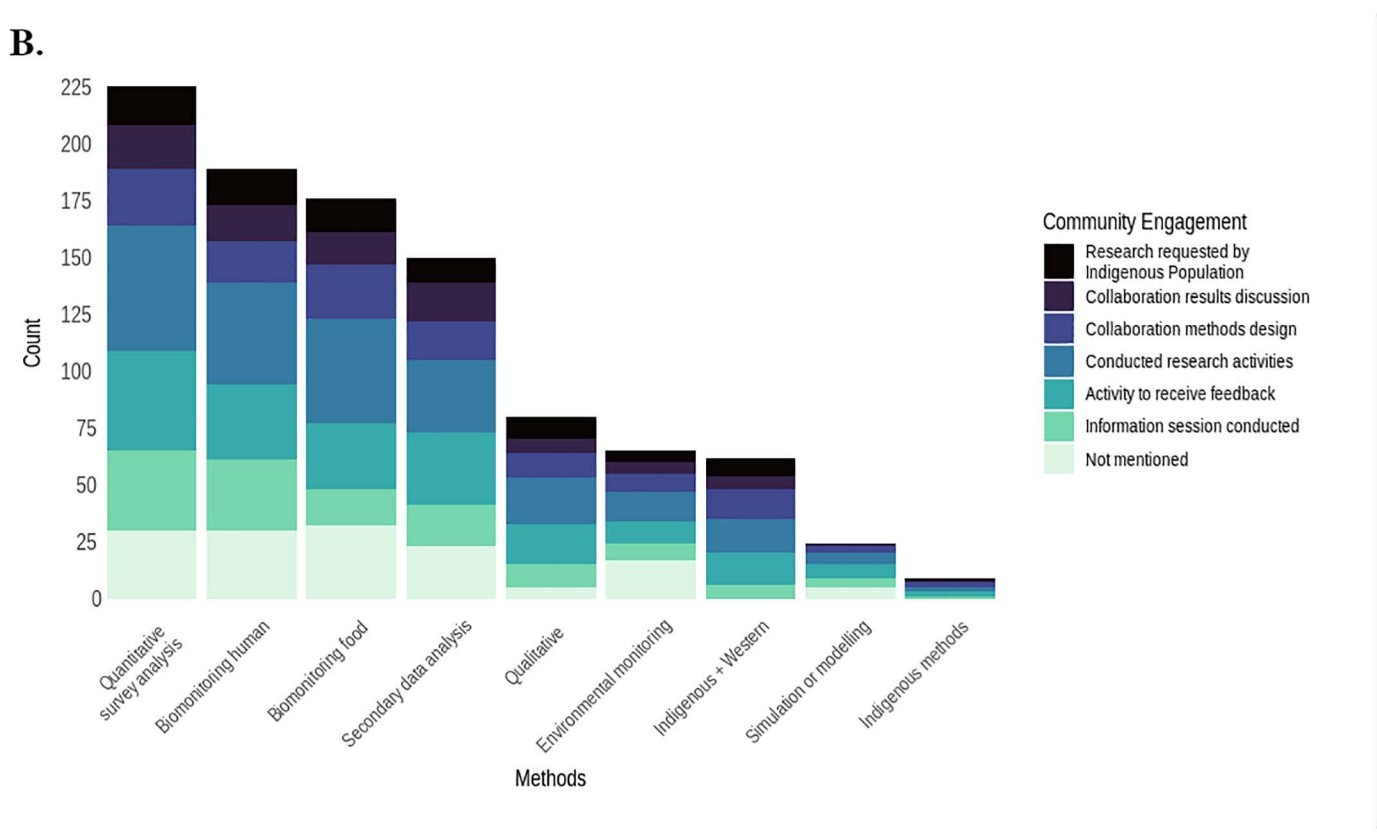

**Fig 2. Community engagement strategy per research method as reported in Indigenous Food systems contamination studies.** A) Proportion B) Frequency (For the "methods" and "community engagement" categories more than one level can be assigned to a single study. Thus, the count is larger than the number of articles included in the review).

strategies for the community to use the study results (Fig 3). The proportions of community results use categories were similar for studies using survey analysis, biological samples, or simulation approaches. The presence of community engagement activities is not exclusive to specific research methods. We identified the six community engagement categories across all research methods, except simulation or modelling methods, for which none of the studies reported to be requested by the community.

In addition, we explored the potential associations between the absence or presence of authors with Indigenous affiliation, both with community engagement and community results use (Figs 4A and 4B). We identified mention of an Indigenous affiliation only in three countries: 37 in Canada, 17 in the United States, and one in Suriname. In the latter, the authors reported conducting an activity to receive community feedback and that the community was using the study results as part of an ongoing broader strategy. In Canada and the United States, studies with an Indigenous affiliation reported a higher proportion of community engagement activities than those without such affiliation. We also observed that studies having an author with an Indigenous affiliation reported a larger proportion of activities to receive community feedback than studies without an Indigenous affiliation. The proportion of studies requested by an Indigenous community was larger in studies with an Indigenous affiliation compared to those without. This was also the case for most community engagement strategies (i.e., collaboration in results discussion, collaboration in methods design, and community members conducting research activities). This could be indicative of an association between Indigenous authorship and community engagement in food systems contamination research.

How study results were reported to be used by the involved communities also differed between articles that were prepared with or without an Indigenous affiliation (Fig 4B). This distinction was more pronounced in the United States, where more than half of the studies with an Indigenous author reported the community using the results to support an ongoing broader strategy or intervention. In contrast, none of the studies without an Indigenous author reported on the current use of the study results by the community. Among studies conducted in Canada, a larger proportion of studies reported on current community use of the results when comparing research undertaken with or without an author with an Indigenous affiliation. Of note, the proportion of articles with an Indigenous affiliation seems to indicate an upward trend across the years included in this review, particularly between 2019 and 2024 (Fig 4C).

## Discussion

The call for environmental health research that promotes community engagement and articulates results with broader community processes is yet to be more widely reflected in the peer-reviewed literature on Indigenous food systems contamination. In this scoping review, we explored study characteristics that could be associated with community-based research. We also reflected on how researchers' reporting practices in scientific journals are responding to Indigenous Peoples' calls for horizontal collaborations. This advocacy for equitable collaborations aims to strengthen study validity, promote solution-based frameworks, improve risk management and communication, and strengthen communities' self-reliance and decision-making [24,55,67–69].

### Country distribution of Indigenous food systems contamination research

Despite Indigenous populations inhabiting ~90 countries globally [18,19], publications about the contamination of Indigenous food systems are concentrated in a few countries. This likely provides an unbalanced picture of the relevance of this

**Fig 3. Community use of study results per research method as reported in Indigenous Food systems contamination studies.** A) Proportion B) Frequency (More than one method can be assigned to a single study. Thus, the count is larger than the number of articles included in the review).

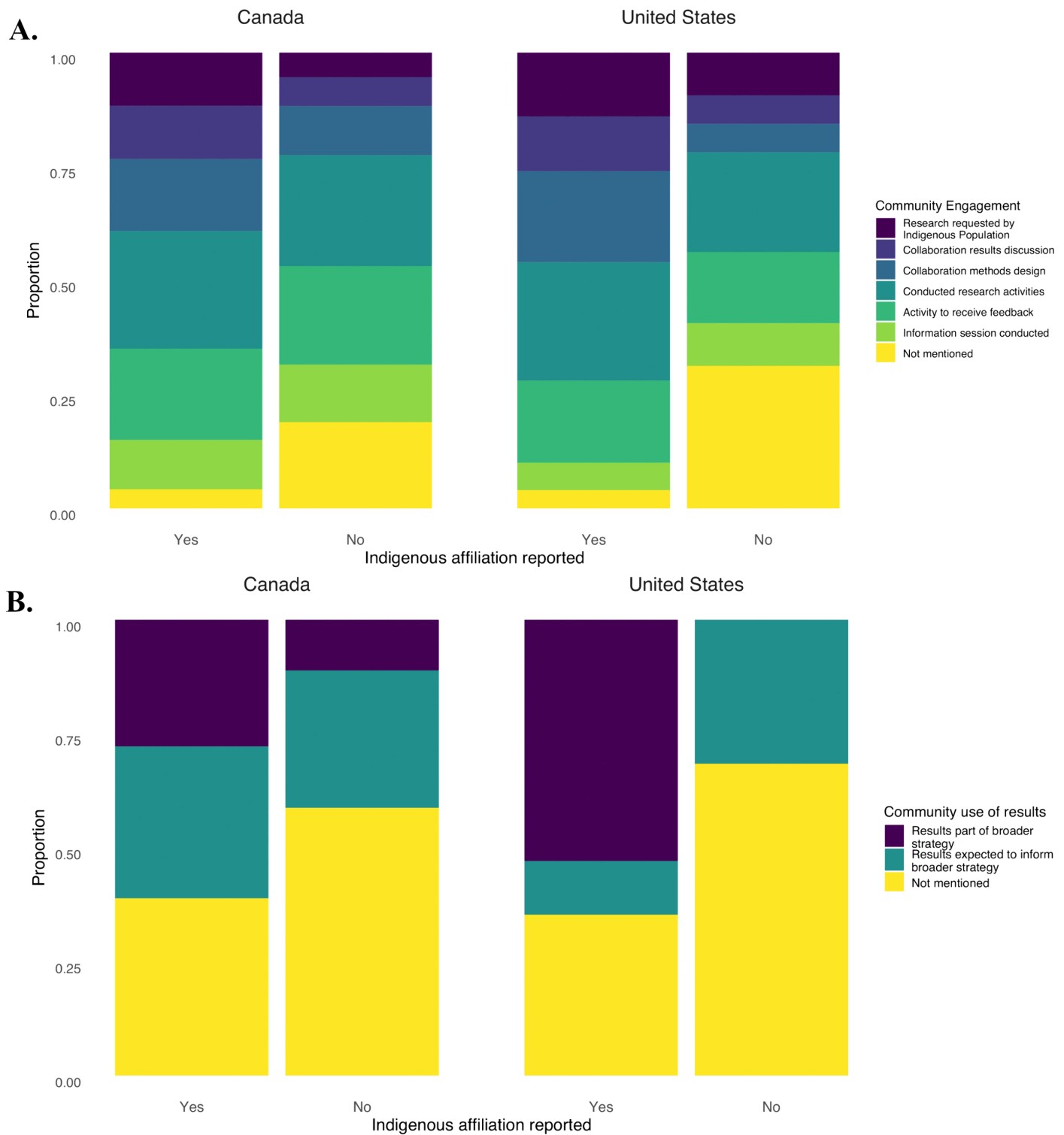

**Fig 4. Indigenous affiliation as reported in Indigenous Food systems contamination studies across A) community engagement strategies, B) community use of studies results, and C) publication year.**

issue globally. The burden of disease from contamination is concentrated in low- and middle-income countries [2], contrasting with most studies included in this review being conducted in high-income countries (Table 2). This is concerning, as some of the greatest exposures to pollution occur outside high-income countries. For example, the bioaccumulation of mercury in Indigenous food systems is a major public health concern [46,70]. Artisanal gold mining is the largest source of mercury emissions, and China, Indonesia, and Colombia are the major documented polluters, with the latter having the highest per capita mercury emissions globally [71,72]. Although these three countries are home to large and diverse Indigenous Peoples, we only identified two studies conducted in Colombia and both among Amazon Indigenous Peoples, which is not illustrative of the vast diversity of Indigenous populations in the country [73].

Most countries with the largest proportion or total population of Indigenous Peoples, as defined by the International Work Group for Indigenous Affairs (IWGIA), are in Asia, Africa, and Oceania [64]. However, we only identified thirteen studies from those regions. This regional bias is partly linked to the global inequality of scientific research funding and production, but other factors are likely to play a role. For example, this discrepancy may be related to limitations in national government and researchers' recognition of these populations, particularly in studies lacking community participation. Furthermore, there is no single definition of Indigenous Peoples, as there are often differing views about the characteristics that define Indigeneity [74,75]. This is further complicated by the ongoing impacts of colonization and the displacement of sovereignty, which continue to impact the social fabric of many Indigenous Peoples [76]. Moreover, depending on the context, communities may choose not to self-identify as Indigenous due to political factors or, in some cases, as a strategy for survival. Thus, the social and political struggle of Indigenous Peoples cannot be separated from the scientific practice of environmental health research and rigorous global assessment of contamination of Indigenous food systems.

However, the accumulated research experiences of Indigenous Peoples in some high-income countries can shed light on collaborative practices that could be implemented in comparable communities elsewhere [48,70]. Moreover, Indigenous Peoples around the globe face similar challenges and often experience an astounding gap in health outcomes compared to non-Indigenous counterparts in their countries [18,77]. For example, the structural racism within healthcare systems in Canada is persistent, and health professionals often lack a deep understanding of the historical and structural violence against Indigenous Peoples [78]. While there is a relatively large and increasing number of environmental health studies conducted in Canada among Indigenous Peoples, the elevated risk of these populations to toxic chemical exposure is persistent.

One factor that may hinder the translation of study results into improving Indigenous environmental health in Canada is that researchers are not adequately including the unique perspective of Indigenous Peoples in their studies or, alternatively, they are not fully communicating community-defined priorities in their publications [14,15]. To prevent cases in which researchers engage in meaningful collaborations with Indigenous Peoples but fail to report these partnerships in scientific publications, editorial policies should adapt to encourage explicit discussions on the role of community engagement in strengthening the rigour, effectiveness, and relevance of studies with a biomonitoring component.

### Bridging Western and Indigenous knowledge through community engagement

Despite the challenges of bridging Indigenous and Western knowledge systems and building relationships with community members [14,79], there is mounting evidence of the benefits of these collaborative efforts on addressing environmental justice issues [49,51,54]. Although there is concern about the increasing negative effects of natural degradation on global food nutrition and security, Indigenous Peoples are known to be disproportionately affected by such adverse environmental changes [80–82]. Indigenous Peoples well-being is closely linked with local natural environments [12]. Thus, the deterioration of ecosystems may impact Indigenous Peoples' health, culture and food systems in a shorter time scale than non-Indigenous populations living in areas with better access to alternative food sources. Moreover, this deeper connection with the natural environment is, above all, a source of unique knowledge that should increasingly become part of applied research in planetary health and other Western scientific fields. Indigenous Peoples' traditions strongly value a collective awareness of the interconnectedness within nature and its inseparability from human well-being [9,51]. In

addition, Indigenous traditional knowledge cannot be unpaired from the community and land [9]. Therefore, a meaningful articulation of Indigenous traditional knowledge and Western sciences requires capturing the particular context of the research.

In addition to the benefits of community engagement and articulating Indigenous Peoples frameworks that better explain the system-level complexity of toxic exposures [56,83,84], a horizontal dialogue to co-develop interventions and research projects could constitute an act towards reconciliation itself [85]. In this review, however, we observed a relatively low proportion of articles reporting including Indigenous knowledge in the study design, and only two studies using Indigenous research methods in addition to Western approaches. Our results are consistent with the findings of Chong and colleagues, who showed that researchers considered the meaningful inclusion of Indigenous knowledge as one of the most difficult challenges in the development of new community-based risk assessment approaches [14].

Indigenous scholars have highlighted that the assumed superiority of Euro-Western-centric knowledge in health research is resulting in an "epistemicide" (i.e., killing, silencing, annihilation, or devaluing of a knowledge system) that homogenizes knowledge and propagates value systems that ignore the interconnectedness of human well-being and nature [10]. Moreover, the meaningful and non-extractive inclusion of Indigenous knowledge and community-level context requires a focus on community engagement that is often lacking in research among Indigenous Peoples [10,14,15]. Despite the persistent challenges to articulate Indigenous and Western knowledge systems, fields like planetary health are progressively recognizing the key contributions of Indigenous knowledge to understanding the complex connection between humans and Nature [69]. As planetary health approaches are incrementally being included in higher education curricula, educators of health professionals and researchers should leverage this opportunity to promote innovative learning approaches that incorporate diverse knowledge systems [86]. This could strengthen the relevance and effectiveness of future risk assessments and studies of Indigenous food systems.

## Community engagement strategies

We explored patterns in the proportion of community engagement strategies across the different methods used in the reviewed studies. The most common strategy was the participation of Indigenous members in conducting research activities, like data collection, followed by activities to receive feedback from the community. Combined, these two strategies presented the largest proportion across most study methods. Overall, 63% of the studies reported some level of community engagement. This figure contrasts with a previous review, which found that only 19% of Indigenous environmental health research conducted between 2000 and 2015 in the Circumpolar North discussed any engagement with the community [15]. This relatively large discrepancy is likely due to differences in the classification of community engagement, and that this review focused on a subset of Indigenous environmental health systems (i.e., food systems contamination). When only considering the three community engagement categories in our study that are comparable to Jones and colleagues' classification (i.e., "Study requested by community", "Results discussion in collaboration with the community", and "Research methods designed in collaboration with the community"), we found that 33% of the studies reported one of these strategies. This percentage is closer to Jones and colleagues' findings and consistent with the increasing trend reported for the 2000–2015 period [15].

In addition, we observed that among the 184 studies that collected human, food, or environmental samples, 34% did not mention a community engagement strategy. Although in some cases the lack of reporting could be caused by journals' editorial constraints that limit authors' ability to report existing community engagement activities, it may also be indicative of common concerning practices in environmental health research. The engagement of communities in the research process is particularly key to improving the sustainability and impact of studies that include a biomonitoring component [54]. Moreover, research among Indigenous Peoples should view community engagement as both a strategy to improve health and reduce hazardous exposures and a rights-based process that can strengthen communities beyond the study or intervention boundaries [57,87].

Approaches centred on the community exposed to pollution are particularly well-positioned to evaluate and manage the impact of contamination while promoting solutions that account for the complex contextual issues that threaten Indigenous Peoples' well-being. This can be achieved by using participatory strategies and community-based projects where community members lead or are meaningfully involved at all levels of the research process. To inform environmental health policies, several participatory community-based biomonitoring studies of traditional food among Indigenous Peoples have indeed been conducted [70,88–90]. In addition to improving the quality of the studies, these community-based initiatives have a number of benefits, such as stimulating community interest in scientific studies, building local capacities, improving risk management and communication, and promoting self-reliance and decision-making at the local level [24,53].

Despite the potential benefits of community-based risk assessments, researchers and community members have reported persistent challenges in their implementation [14]. Partnerships between academia and communities could benefit from reflecting on better strategies to leverage solutions to contamination issues [91]. Previous studies have found that research directed by communities experiencing environmental injustice is more likely to result in action than research led by academic institutions [91]. Thus, institutions of higher education should use their power and prestige to amplify communities' voices, while expanding opportunities to play a supportive, rather than leading, role in environmental health research [69,91].

Although not exclusive to qualitative methods, community engagement strategies were more commonly reported in studies that included them. This is likely related to the broad discussion in qualitative research on strategies to promote participation. Indeed, the validation of study results with the community is a criterion for assessing rigour in qualitative studies. Our results are consistent with Jones and colleagues' conclusions on environment-related Indigenous health research in the Circumpolar North, reinforcing the authors' call for further development of community engagement strategies in quantitative methods [15]. Hayward and colleagues discussed the dominance of qualitative methodologies in Indigenous research and the need for Indigenous quantitative methods to be expanded and strengthened [92]. They suggested a "strength-based" approach where the quantitative data analysis process acknowledges the positionality of the researchers and is consistent with community values and principles. This aligns with calls from Indigenous scholars for enforcing explicit positionality statements in research articles reporting on studies among Indigenous Peoples [10,93].

In addition, the inclusion of qualitative components and mixed methods should be considered in tandem with future quantitative research due to the importance of understanding contextual factors (e.g., food preferences and cultural values of traditional foods), including the reasons behind decisions that affect the environmental justice of Indigenous Peoples [94].

In this review, we found that ~17% of studies measuring contamination in human samples reported to have been requested by Indigenous Peoples. Moreover, a third of such studies did not mention any community engagement strategy. Due to the potential negative influence of human biomonitoring results on community members' practices and perceptions, incorporating community engagement activities throughout the research process can strengthen locally relevant risk management and communication strategies [57]. Thus, we were expecting a relatively large proportion of community engagement strategies reported in the studies that collected human samples. However, this is not reflected in our data. Similar challenges can arise in food biomonitoring studies, as it can increase health concerns from consuming traditional foods [56]. The communication of biomonitoring studies results is a major challenge, including cases where low contamination levels were found, because it has the potential to shape community risk perception of exposure and unintentionally discourage beneficial practices, such as Indigenous food harvesting. Moreover, there is evidence that researchers consider risk communication the most challenging part of risk assessments among Indigenous and local populations [14]. However, most of these challenges can be addressed by developing equitable partnership with communities where researchers support risk evaluation but local and regional organizations lead risk management and communication, such as is the case in Nunavik [24].

Strengthening local and regional sovereignty over risk management and communication is key to reducing exposure to high-risk food due to contamination with toxic chemicals while balancing the health benefits of traditional food harvesting. National all-encompassing consumption advisories may be appropriate for populations with access to other healthy food sources, but promote unintended reduction of traditional foods. Such displacement of food preferences may heighten the risk of disease in Indigenous Peoples by increasing the prevalence of unhealthy diets and decreasing the benefits of traditional food harvesting [28,45]. Thus, to minimize the risk of toxic contamination and strengthen the benefits of traditional food systems in Indigenous Peoples, it is crucial to incorporate adequate and culturally sensitive risk assessment, management, and communication strategies [57,88,95].

**Beyond documenting damage: Community use of study results and Indigenous authorship**

In connection with the benefits mentioned earlier, community engagement can potentially expand the focus of environmental health scientific inquiry beyond solely documenting damage. Indigenous scholars have advocated suspending "damage-centred" research and reframing such projects as "desire-centred" inquiries [55]. This approach can produce analyses that challenge assumptions with regard to responsibilities, accountability and opportunities, and aid in dismantling the structures that position these communities as damaged [55]. Thus, the research process should not only respond to the requirements of non-Indigenous researchers but also factor in how the results are articulated and used in broader strategies to restore and promote traditional food systems as well as intergenerational knowledge transmission about traditional food harvesting, preparation, consumption and health and well-being properties.

In this review, we found that most studies that collected human, food, or environmental samples did not mention if the results were articulated with other present or future projects or interventions to control exposure, advocate for environmental justice, and/or improve environmental quality. When looking at studies that conducted qualitative research or articulated Indigenous and Western methods, the proportion of studies discussing how the community was using the results was relatively larger than for studies using quantitative methods exclusively. This is problematic because the lack of discussion across scientific fields and communities of practice may confine the dialogues on the inclusion of environmental justice and community-based participatory research to those already familiarized with the benefits and importance of these approaches.

Although these results may, in fact, represent a concerning pattern in quantitative research, the limited discussion of community use of study results in research articles is likely partly associated with different scientific writing practices across research paradigms. Journals that publish qualitative research may expect and encourage researchers to report and discuss more broadly how the study was integrated into other social processes beyond the boundaries of the study design. This practice should be expanded and adapted to journals focused on quantitative studies, encouraging authors who are part of equitable partnerships with Indigenous Peoples to reflect and recognize the contribution of community engagement to the research and use of the study results. Thus, we highlight recommendations of Indigenous scholars for academic journals to require studies to explicitly report how the affected community was involved in all stages of the research process [10]. Several studies in this review included a brief discussion on these issues in journals focused on quantitative environmental health and exposure sciences. Moreover, journals like Arctic Science accept contribution recognition and author's positionality sections before the introduction, where Indigenous community members' roles throughout the study can be detailed [96].

Indigenous Peoples have long advocated for self-determination and engagement with research conducted in their communities. They have been at the forefront of strategies to improve collaborations with non-Indigenous researchers. For example, the First Nations principles of ownership, control, access, and possession (OCAP®), were created as a guiding tool for Indigenous Peoples and researchers to support Indigenous information governance and data sovereignty [97,98]. Thus, the scientific community must reciprocate these efforts and normalize conversations in scientific journals about community participation and the use of study results as part of broader acts of reconciliation with Indigenous Peoples [93]. These conversations in academic journals could provide valuable

insights into pressing issues such as the uncritical expansion of policies mandating data openness, without considering how they can create mistrust among Indigenous partners and further exclude these populations from participating in research [99,100].

Finally, from the evidence of authors with Indigenous affiliation in Canada, the United States, and Suriname, we observed a potential pattern where community engagement and community use of the results were more likely in studies with authors affiliated with Indigenous organizations or institutions. This finding underlines the importance of including Indigenous authors in the project and its publication. Further analyses should be conducted to understand if having an Indigenous author influences community engagement practices and community use of study results or if the association mostly flows in the opposite direction. In addition, we identified an increasing trend in the proportion of articles with Indigenous authorship across the study period, particularly from 2019 to 2024, which is consistent with previous reviews [15]. The trends and effects of Indigenous authorship warrant future monitoring, given the revitalized calls from Indigenous scholars to expand the concept of "nothing about us without us" to academic publications [10,93], and the emerging strategies by some journals to promote community engagement of Indigenous Peoples either as authors or through meaningful roles in the research process.

### Limitations and strengths

This study has several limitations. Our review did not include grey literature, which may be particularly important in identifying assessments conducted by public and private institutions, including Indigenous ones. Although this was beyond our focus on peer-reviewed research, there is room to expand the analysis to understand further how assessments of Indigenous food contamination are conducted beyond academia. More peer-reviewed and grey literature studies could be identified by searching for cited materials and publications or citing references and requesting access to reports from organizations that have monitored contamination levels.

Additionally, researchers who promote equitable partnerships with Indigenous Peoples may not be reporting on strategies to support community engagement and the use of the study results by the community in their scientific publications. Thus, our classification may not comprehensively reflect the characteristics of the research process, but the constraints established by editorial policies. Moreover, there could be cases where the limited information provided in articles regarding the role of the community in the research process is due to the community's preference. However, in addition to the intrinsic value of evaluating researchers' practices and discussions in scientific publications, we believe that our findings are indicative of general trends about the nature of current collaborations between environmental health researchers and Indigenous Peoples. Further research should be conducted to assess the quality of the community engagement activities and the use of study results beyond what is reported in scientific publications.

Our search may not have captured relevant research that either did not refer to the population or food system as Indigenous, or that was conducted in a language other than English. Despite not restricting the language of publication, we only searched for the terms in English, which could bias the countries included in the sample. An additional bias of the countries included in the study may emerge from the fact that some traditional communities do not identify as Indigenous or are situated in countries that do not recognize Indigenous Peoples. Although we expanded our search keywords following recommendations to capture a wide range of terms used for Indigenous Peoples, our sample is likely biased towards countries where Indigeneity is experienced in a particular way and Indigenous Peoples are officially acknowledged and recognized. Further research accounting for particular regions or contexts should be conducted.

Finally, our approach to identifying Indigenous affiliation may lead to some misclassification. For example, an Indigenous collaborator could choose not to disclose their affiliation in the paper or choose not to be listed as an author due to the added workload of co-authoring a publication. In addition, some journals may have guidelines that discourage adding an Indigenous affiliation, such as limits to reporting multiple affiliations and not allowing group authorship. Despite these

limitations, this scoping review provided a systematic approach to include a broad range of articles exploring the contamination of Indigenous food systems. We followed PRISMA protocols and leveraged the flexibility of scoping reviews to discuss our research experience in the context of literature in the field.

## Conclusion

The increasing appreciation in academia of Indigenous and traditional knowledge systems must continue moving beyond an intellectual exercise and translate into a reduction of the existing health disparities between Indigenous and non-Indigenous counterparts. Beyond the benefits of more equitable collaboration and integrated approaches that better explain the complexity of toxic exposures in Indigenous Peoples, co-developing interventions and research projects to evaluate and manage risk while considering nutritional benefits and cultural importance of traditional foods could constitute an act toward reconciliation itself. Further articulation of traditional knowledge into existing population and planetary health frameworks is crucial to better adapt environmental health interventions and research projects to the reality of the affected communities. Research on toxic exposures in historically oppressed populations should align with the voices of Indigenous scholars advocating to reframe environmental health research to also capture the hope, wisdom and stewardship of communities. One starting point is exploring alternative methods to capture the complexity of contamination issues in traditional food systems and building strategies to work side-by-side with Indigenous Peoples throughout the research process. Moreover, researchers working in solidarity with Indigenous Peoples are responsible for building meaningful long-term collaborations that recognize the sovereign aspirations and leadership of Indigenous Peoples in implementing their visions for a healthy and sustainable future.

## Supporting information

**S1 Table. Articles included in the review organized by first author last name.**
(DOCX)

**S2 Table. Data extracted from each individual article included in the review organized by first author last name.**
(DOCX)

**S3 Table. Indigenous Populations in the studies included in the review as named in the article, sorted by frequency.**
(DOCX)

## Acknowledgments

We would like to thank Ursula Ellis, librarian at the University of British Columbia, for their valuable guidance in developing the systematic review strategy. We also extend our gratitude to Indigenous leaders for their continued advocacy and collaboration in strengthening environmental exposure assessments and management programs, which are centered on community engagement values.

## Author contributions

**Conceptualization:** Federico Andrade-Rivas, Hallah Kassem, Chenoa Cassidy-Matthews, Matthew Little, Melanie Lemire, Annalee Yassi, Jerry Spiegel.

**Data curation:** Federico Andrade-Rivas, Hallah Kassem.

**Formal analysis:** Federico Andrade-Rivas, Hallah Kassem, Kira Mok.

**Investigation:** Federico Andrade-Rivas, Hallah Kassem, Kira Mok, Chenoa Cassidy-Matthews, Jerry Spiegel.

**Methodology:** Federico Andrade-Rivas, Hallah Kassem, Chenoa Cassidy-Matthews, Jerry Spiegel.

**Project administration:** Federico Andrade-Rivas.

**Supervision:** Jerry Spiegel.

**Validation:** Federico Andrade-Rivas, Hallah Kassem, Kira Mok, Chenoa Cassidy-Matthews, Matthew Little, Melanie Lemire, Annalee Yassi.

**Visualization:** Federico Andrade-Rivas.

**Writing – original draft:** Federico Andrade-Rivas.

**Writing – review & editing:** Federico Andrade-Rivas, Hallah Kassem, Kira Mok, Chenoa Cassidy-Matthews, Matthew Little, Melanie Lemire, Annalee Yassi, Jerry Spiegel.

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
