## [Decision Letter · Decision Letter 0]

16 Jul 2025

Dear Dr. Spiegel,

We look forward to receiving your revised manuscript.

Kind regards,

Jenilee Gobin

Academic Editor

PLOS ONE

Journal Requirements:

2. We note that your Data Availability Statement is currently as follows: All relevant data are within the manuscript and in Supporting Information files.

3. Please upload a copy of Supporting Information 1 and 2 which you refer to in your text on page 37.

Reviewers' comments:

Reviewer's Responses to Questions

**Comments to the Author**

1. Is the manuscript technically sound, and do the data support the conclusions?

Reviewer #1: Yes

Reviewer #2: Yes

2. Has the statistical analysis been performed appropriately and rigorously?

Reviewer #1: Yes

Reviewer #2: Yes

3. Have the authors made all data underlying the findings in their manuscript fully available?

Reviewer #1: No

Reviewer #2: Yes

4. Is the manuscript presented in an intelligible fashion and written in standard English?

Reviewer #1: Yes

Reviewer #2: Yes

Reviewer #1: This is an interesting piece of work. It’s clearly written and engages with a critical set of questions around Indigenous engagement in environmental health research. I appreciate the effort to bring visibility to issues surrounding decolonizing research practices in a structured way. That said, there are several areas that need revision to strengthen the manuscript and ensure the conclusions are well-supported by the data.

Areas for Revision:

1. The absence of the full list of included studies (cited as S1 and S2 Tables) is a significant omission; it was impossible to assess the included literature and assess it. Table S3 was missing as well.

2. The manuscript includes broad assertions not always supported by the data. For example:

"Despite the increasing number of environmental health studies conducted in Canada among Indigenous populations, the elevated risk... is persistent and researchers are not fully communicating community-defined priorities in their publications."

I’ve interpreted this sentence as implying a causal link between persistent exposure and researchers’ failure to reflect community priorities, while generalizing about the entire field. Such claims overstate the evidence presented in a scoping review. It is important to acknowledge that academic publications are often constrained by journal formats, word limits, or editorial policies that may limit authors’ ability to include detailed community perspectives/priorities. Some researchers may, in fact, be aligning closely with community priorities in practice even if this is not fully reflected in published outputs. Claims throughout the manuscript should be more cautiously framed and more closely tied to the data presented to improve analytical rigour and credibility, or you run the risk of making generalized claims that are not necessarily true.

3. Only two studies are noted as using Indigenous research methods, but the manuscript does not describe these approaches/ studies (this is mentioned 2-3 times). A brief explanation would help clarify what qualifies as Indigenous research methodology in this context.

4. Tables 2 and 3 contain a lot of useful data, but the main text mostly lists or repeats those numbers without interpreting what they mean or highlighting key patterns. The results section would benefit from clearer synthesis, such as drawing out trends, contrasts, or implications from the tables, rather than just presenting counts. This is one my biggest concerns, you have presented interesting findings, but you need to give the reader more of a “so what?” or “why should I care as a researcher, policy maker, community member, etc.?”

----

Editing suggestions

The writing is clear, but some sentences are quite long or dense, making them harder to follow than they need to be. You’re presenting the state of the art, be clear and concise!

Introduction

Line 76: “The growing complexity of environmental contamination, including new chemicals and mixtures and multi-level pollution drivers…”

Suggested: “…including emerging chemicals, mixtures, and multi-level drivers of pollution…”

Line 83: “This is the case of environmental health research in the Circumpolar North…”

Suggested: “This is particularly evident in environmental health research in the Circumpolar North…”

Line 104: “Unfortunately, despite the increasing interest in fostering collaborations with Indigenous Peoples…”

Suggested: “Although interest in fostering collaboration is increasing…”

Methods

Line 178: “Cited or citing references were not examined, and additional studies were not sought through other strategies.”

Suggested: “We did not examine cited or citing references, nor did we search grey literature or consult external sources.”

Line 196: “We verified if organizations were governed by Indigenous Peoples and classified those that explicitly mention their connection…”

Suggested: “We verified whether the organizations were governed by Indigenous Peoples…”

Results

Line 220: “This number was based on the names that authors assigned in their articles…” [I found this sentence a bit vague, starting on line 219.]

Suggested: “This count reflects the names used by authors in the original studies…”

Line 252: “We also explored the activities that were reported to have been carried out…”

Suggested: “We also examined reported community engagement activities…”

Line 281: “Around 60% of articles did not provide information on community use of study results…”

Suggested: “Approximately 60% of articles did not report…”

Discussion

Line 365: “Despite the challenges of articulating contrasting knowledge systems…”

Suggested: “Despite the challenges of bridging Indigenous and Western knowledge systems…”

Line 431: “Community engagement strategies are not exclusive to qualitative methods…”

Suggested: “Although not exclusive to qualitative methods, community engagement strategies were more commonly reported in studies that included them.”

Conclusion

Line 565: “Beyond the benefits of a horizontal dialogue in incorporating analytical frameworks…”

Suggested: “Beyond the benefits of more equitable collaboration and integrated approaches…”

Line 571: “Research on toxic exposures among historically oppressed populations needs to be in line with the voices of Indigenous scholars…”

Suggested: “Research on toxic exposures in historically marginalized populations should align with the priorities and perspectives of Indigenous scholars…”

Finally, it would help to standardize terminology, for example terms like “Indigenous Peoples” and “Indigenous populations” and/ or “Indigenous communities” seem to be used interchangeably, and a consistent choice would improve clarity.

With these revisions, the manuscript will be well-positioned to make a meaningful and lasting contributions to the literature on Indigenous health, environmental contamination, and improve research relationships with Indigenous partners.

Reviewer #2: Thank you for your work in this area. A comprehensive systematic search of the literature following a scoping review process was conducted to identify and describe reported practices for engaging Indigenous communities in research focused on contaminated food systems. The manuscript is well-written but would benefit from a few revisions to strengthen the methodological approach described and the implications of the scoping review findings.

- Currently, the method section is described at a high level. The authors may consider providing further details on their search strategy, eligibility criteria, data extraction, classification, and analysis. Additionally, the authors may consider citing the process they followed for the systematic scoping review. Providing details on how articles were screened and selected for data extraction would help to highlight the systematic steps the research team took to ensure rigour and reduce bias in the scoping review process.

-Lines 169-171 read as an objective statement that could be moved to the introduction. Your methods section could then lead with a description of the search strategy.

- Inclusion/exclusion criteria – Providing further information on your inclusion criteria, including geographic context, type of source (could be more specific to indicate primary research?), would help to clarify the scope of the review to the reader and set up how your results are currently organized and presented.

- Data analysis – more information is needed here. The authors indicate using R statistical software, but don’t describe what it was used for and why.

-Information on the role of co-authors in either developing the search strategy, eligibility criteria, screening and article selection could be provided. For example, in lines 190-191, you could include the initials of the two independent reviewers.

As the authors indicate that their study as a scoping review, I suggest the authors check their paper against the PRISMA Extension for Scoping Reviews (PRISMA-ScR): Tricco, A. C., Lillie, E., Zarin, W., O'Brien, K. K., Colquhoun, H., Levac, D., Moher, D., Peters, M. D. J., Horsley, T., Weeks, L., Hempel, S., Akl, E. A., Chang, C., McGowan, J., Stewart, L., Hartling, L., Aldcroft, A., Wilson, M. G., Garritty, C., Lewin, S., … Straus, S. E. (2018). PRISMA Extension for Scoping Reviews (PRISMA-ScR): Checklist and Explanation. Annals of internal medicine, 169(7), 467–473. https://doi.org/10.7326/M18-0850

**Do you want your identity to be public for this peer review?** For information about this choice, including consent withdrawal, please see our Privacy Policy

Reviewer #1: No

Reviewer #2: No

---

## [Author Response · Author response to Decision Letter 1]

29 Sep 2025

Our detailed response can be found in the "Response2Reviewers_Sept29.docx" file.

---

## [Decision Letter · Decision Letter 1]

26 Oct 2025

Community engagement in Indigenous food systems contamination studies: A systematic scoping review

PONE-D-25-27598R1

Dear Dr. Spiegel,

We’re pleased to inform you that your manuscript has been judged scientifically suitable for publication and will be formally accepted for publication once it meets all outstanding technical requirements.

Kind regards,

Jenilee Gobin

Academic Editor

PLOS ONE

Additional Editor Comments (optional):

Reviewers' comments:

Reviewer's Responses to Questions

**Comments to the Author**

Reviewer #1: All comments have been addressed

Reviewer #2: All comments have been addressed

2. Is the manuscript technically sound, and do the data support the conclusions?

Reviewer #1: Yes

Reviewer #2: Yes

3. Has the statistical analysis been performed appropriately and rigorously?

Reviewer #1: Yes

Reviewer #2: Yes

4. Have the authors made all data underlying the findings in their manuscript fully available?

Reviewer #1: Yes

Reviewer #2: Yes

5. Is the manuscript presented in an intelligible fashion and written in standard English?

Reviewer #1: (No Response)

Reviewer #2: Yes

Reviewer #1: Thank you for engaging with my comments in a thoughtful manner; a great piece of work for our field.

Reviewer #2: Thank you for your contributions to the literature on this topic! Congratulations to all the authors.

**Do you want your identity to be public for this peer review?** For information about this choice, including consent withdrawal, please see our Privacy Policy

Reviewer #1: No

Reviewer #2: No

---

## [Editor Report · Acceptance letter]

PONE-D-25-27598R1

PLOS ONE

Dear Dr. Spiegel,

I'm pleased to inform you that your manuscript has been deemed suitable for publication in PLOS ONE. Congratulations! Your manuscript is now being handed over to our production team.

Kind regards,

on behalf of

Dr. Jenilee Gobin

Academic Editor

PLOS ONE